# Effort produces after-effects costly for others but valued for self

Ya Zheng[1,2]*[†], Rumeng Tang[3,4][†]

[1]Department of Psychology, Guangzhou University, Guangzhou, China; [2]Center for Reward and Social Cognition, School of Education, Guangzhou University, Guangzhou, China; [3]Guangdong Provincial Key Laboratory of Social Cognitive Neuroscience and Mental Health and Department of Psychology, Sun Yat-sen University, Guangzhou, China; [4]Department of Psychology, Dalian Medical University, Dalian, China

## eLife Assessment

The findings in this paper provide **solid** support for a hypothesis that has **valuable** implications at the intersection of value-based and social decision-making. The findings suggest that the brain processes rewards received for effort differently when they are earned for themselves versus someone else.

*For correspondence:
zhengya1982@gmail.com

[†]These authors contributed equally to this work

Competing interest: The authors declare that no competing interests exist.

**Abstract** Engaging in prosocial behavior requires effort, yet people are often averse to exerting effort for others' benefit. However, it remains unclear how effort exertion affects subsequent reward evaluation during prosocial acts. Here, we combined high-temporal-resolution electroencephalography with a paradigm that independently manipulated physical effort and monetary reward for self and others to elucidate the neural mechanisms underlying the reward after-effect of prosocial effort expenditure. We found dissociable reward after-effects for self-benefiting and other-benefiting effort. For self-benefiting rewards, the reward positivity (RewP) increased with effort demand, suggesting an effort-enhancement effect. In contrast, for other-benefiting rewards, the RewP decreased as effort increased, demonstrating an effort-discounting effect. Critically, this dissociation was contingent upon high reward magnitude and modulated by individual differences in effort discounting, yet remained distinct from performance evaluation. Our findings reveal distinct neural computations for self- and other-benefiting efforts, offering new insights into how prior effort expenditure shapes reward evaluation during prosocial behavior.

## Introduction

Prosocial behaviors are acts that benefit others and often involve some personal costs (*Bierhoff, 2002*). These actions are crucial for promoting individual physical (*Post, 2005*) and mental (*Raposa et al., 2016*) well-being and serve as a powerful force enhancing social cohesiveness and group bonding (*Fehr and Fischbacher, 2003*). According to the cost-benefit framework (*Contreras-Huerta et al., 2020*), people help others either because they are highly sensitive to others' welfare or because they are less sensitive to their own costs. However, prior studies have predominantly manipulated only financial costs (*Engel, 2011*), ignoring the most common cost type: effort, a non-contextual factor that represents the physical or cognitive resources required for goal-directed behavior (*Shenhav et al., 2017*). Indeed, prosocial behavior in everyday life requires the investment of varying amounts of effort, whether helping a colleague proofread a paper or holding an elevator for a stranger. In this study, we examined how effort exertion affects subsequent reward evaluation during prosocial acts.

Effort is typically considered costly and aversive. All else being equal, people usually follow a 'law of less work' (*Hull, 1943*) and prefer lower over higher task demands (*Kool et al., 2010*), manifesting as effort discounting (*Westbrook et al., 2013*). Due to its inherent aversiveness, effort serves as an ideal proxy for cost in studying prosocial behavior. Recent studies have characterized the psychological and neurobiological mechanisms underlying prosocial effort. In these studies, participants repeatedly decide whether to invest effort to gain financial rewards for themselves or others. A consistent finding is that participants are less willing to exert effort to benefit others than themselves, specifically when the required effort is high. This prosocial apathy is observed for both physical effort (*Lockwood et al., 2017*) and cognitive effort (*Depow et al., 2022*). It is modulated by acute stress (*Forbes et al., 2024*), aging (*Lockwood et al., 2021*), and individual differences in broad affective traits (*Contreras-Huerta et al., 2022*). Moreover, prosocial acts are invigorated to a lesser degree during effort exertion than identical self-benefiting ones (*Lockwood et al., 2017*; *Lockwood et al., 2022*). Neuroimaging and brain lesion studies have revealed that prosocial effort costs are tracked by neural activity in the anterior cingulate gyrus, anterior insula, and ventromedial prefrontal cortex (*Forbes et al., 2024*; *Lockwood et al., 2024*; *Lockwood et al., 2022*). While these studies have focused on how people choose to exert effort during decision-making and energize their actions during effort exertion, they have largely ignored another important aspect of effort expenditure: the reward after-effect of effort expenditure.

The reward after-effect of effort expenditure refers to a temporary tuning toward or amplification of reward-related motivation following effort exertion (*Inzlicht et al., 2018*; *Kelley et al., 2019*). Specifically, people assign more value to rewards they have put effort into and are more reluctant to share monetary rewards earned through greater effort (*Arkes et al., 1994*; *Norton et al., 2012*). The effort-adds-value phenomenon is observed across species (*Lydall et al., 2010*; *Pompilio et al., 2006*), suggesting that it is biologically hard-wired. Echoing these behavioral findings, neuroimaging evidence has demonstrated that prior effort investment increases brain activity in reward-related neural circuits, including the anterior cingulate cortex, orbitofrontal cortex, and ventral striatum (*Dobryakova et al., 2017*; *Hernandez Lallement et al., 2014*; *Wagner et al., 2013*). Further evidence comes from studies focusing on the reward positivity (RewP) of the event-related potential (ERP) component. The RewP, a reliable neural signature of reward sensitivity, has its neural sources in the anterior cingulate cortex (*Hauser et al., 2014*). Some studies found that RewP amplitude increased following the exertion of high-effort versus low-effort behavior for reward feedback (*Bogdanov et al., 2022*; *Harmon-Jones et al., 2024*; *Ma et al., 2014*; *Umemoto et al., 2023*; *Yi et al., 2020*). While this framework has been used extensively to understand how effort expenditure influences reward processing during self-benefiting behaviors, surprisingly, it has not yet been applied to understand the reward after-effect of effort expenditure during prosocial acts.

In this study, we aimed to investigate the reward after-effect of prosocial effort, focusing on the RewP elicited during reward evaluation after effort expenditure. Previous studies examining how people vicariously process others' rewards suggest a critical role of the RewP in prosocial behavior (*Kwak et al., 2020*; *San Martín et al., 2016*). In these experiments, participants performed simple gambling or speeding reaction-time tasks to win monetary rewards for themselves and others. A consistent finding is that the RewP is attenuated for reward feedback benefiting others relative to oneself, including charity programs (*San Martín et al., 2016*), beneficiaries with closer social distance (*Kwak et al., 2020*), or anonymous individuals (*Li et al., 2022*). However, rewards in these studies were earned with minimal effort. Crucially, previous research has never considered the impact of prior effort costs.

To address this issue, we recorded the RewP in a prosocial effort task where participants exerted varying levels of physical effort to earn monetary rewards for themselves or an anonymous other person. We hypothesized an effort-discounting effect on the RewP when exerting effort for others, indicated by a reduced RewP as required effort increased. Conversely, we expected an effort-enhancement effect on the RewP when participants put in effort for themselves, shown by a more positive RewP as required effort increased. Further, we predicted that this dissociation would be contingent upon reward system activation. To provide a comprehensive characterization of prosocial effort, we subsequently examined participants' decision-making tendencies in a prosocial decision-making task where they chose to exert effort to benefit either themselves or others. We hypothesized

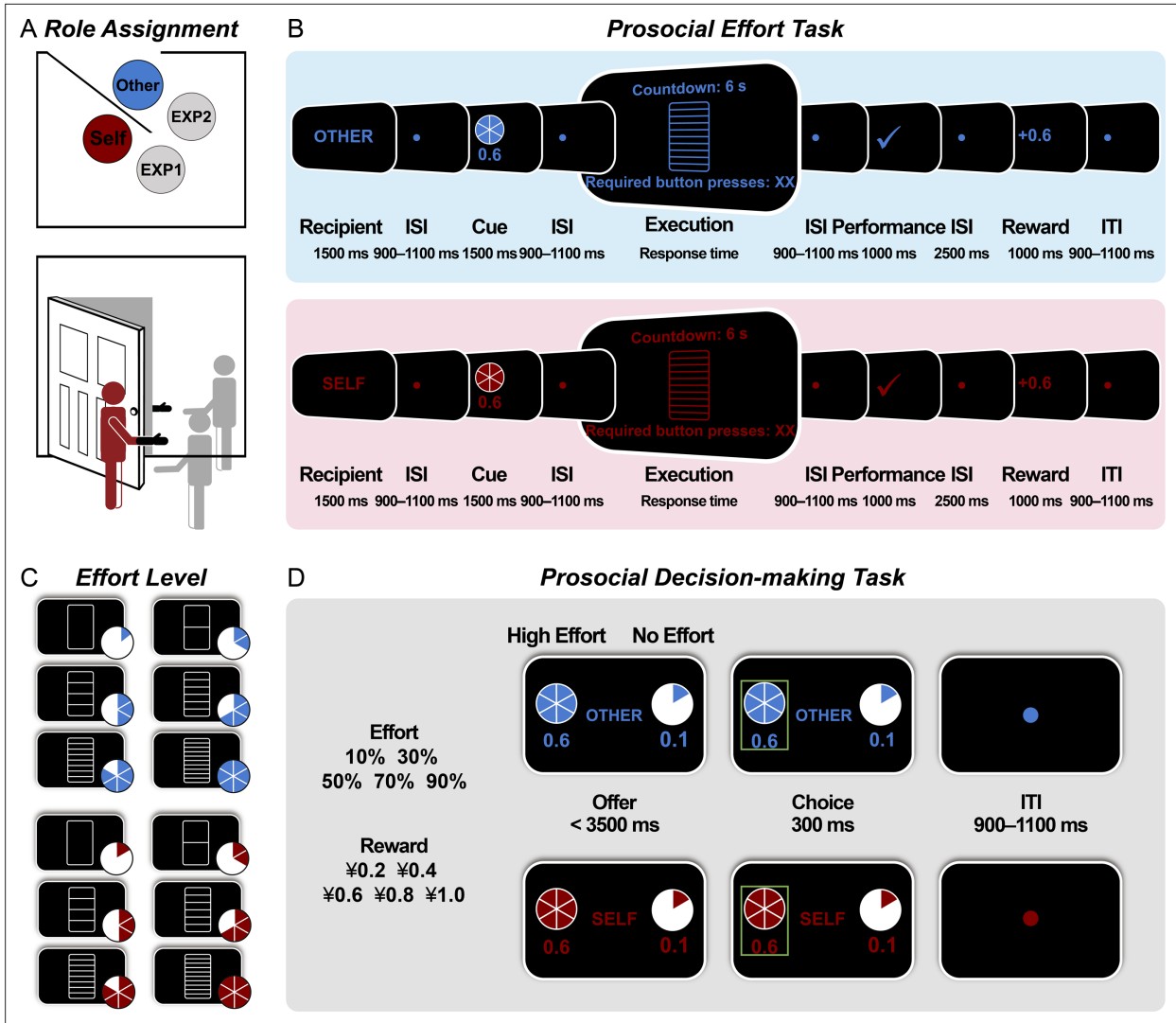

**Figure 1.** Experimental tasks. (**A**) The role assignment task. Participants were introduced to another anonymous person and designated as a decider to invest physical effort for monetary rewards for themselves and others. (**B**) The prosocial effort task. Participants exerted physical effort (five levels: 10%, 30%, 50%, 70%, or 90% of the participant's calibrated maximum effort, shown as bars 2–6 in panel C) to earn a potential reward of varying amounts (¥0.2, ¥0.4, ¥0.6, ¥0.8, or ¥1.0) for themselves and others. Successful effort had a 50% chance of yielding a reward. (**C**) Effort levels. The physical task required participants to rapidly press a button with their nondominant pinky finger within 6000 ms. Each effort level was visualized as the height of a vertical bar. The leftmost blank bar (bar 1) indicated no effort and was used only in the baseline option of the prosocial decision-making task. (**D**) The prosocial decision-making task. Participants chose between a high-effort option (more effort for a larger reward) and a baseline option (no effort for a smaller reward). ISI = interstimulus interval; ITI = intertrial interval.

that participants would exhibit reduced willingness to invest effort for others (i.e. prosocial apathy), and this behavioral tendency would modulate the neural after-effects of effort exertion.

## Results

In this study, participants completed a role assignment task (*Figure 1A*) to become the decider in two subsequent prosocial tasks: a prosocial effort task and a prosocial decision-making task. In the prosocial effort task (*Figure 1B*), participants exerted five levels of physical effort (2–6 levels; *Figure 1C*) to earn rewards of varying amounts (¥0.2, ¥0.4, ¥0.6, ¥0.8, or ¥1.0) for themselves or an anonymous other person. The five effort levels were fully crossed with the five reward magnitudes, creating 25 unique combinations. Following the effort task, participants completed a prosocial decision-making task (*Figure 1D*) where they chose between a high-effort option (more effort for a larger reward) and a

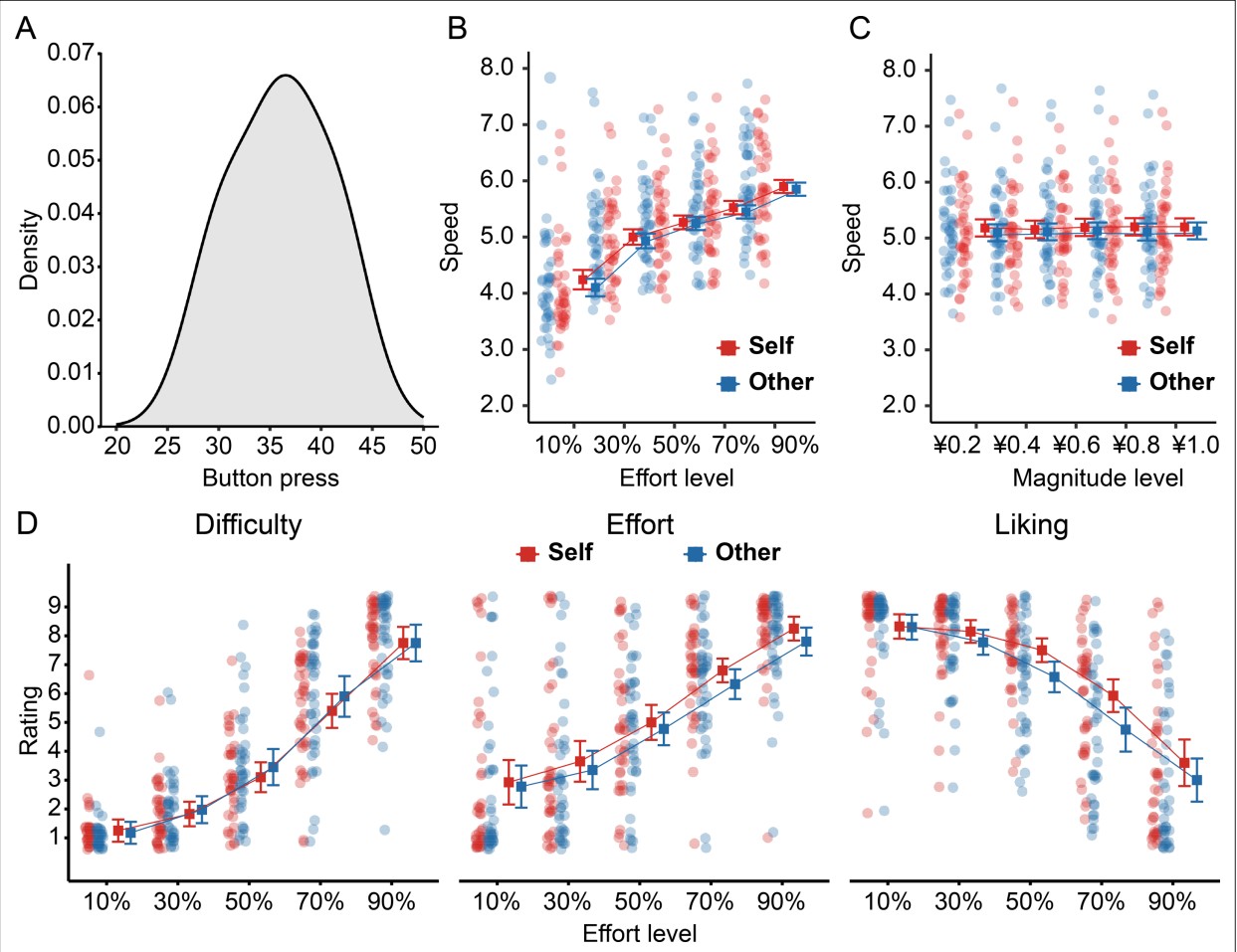

**Figure 2.** Behavioral and rating results of the prosocial effort task. (**A**) Distribution of the maximum effort level (i.e. the average button-press count across three 6000 ms trials) across participants. (**B–C**) Response speed (button presses per second) data. Participants responded faster for themselves than for others. Response speed also increased as effort demands increased. (**D**) Rating data. Participants felt less effort and more disliking when exerting effort for others than for themselves. Error bars represent the within-subject standard error of the mean (n = 40).

baseline option (no effort for a smaller reward). The high-effort options comprised the same 25 unique effort-reward combinations presented in the prosocial effort task. In both tasks, half of the trials benefited participants themselves, while the other half benefited others. We measured participants' neural responses to rewards obtained following effort exertion for themselves and others in the prosocial effort task, as well as their willingness to exert prosocial effort in the decision-making task. Additionally, to validate the manipulation, we collected participants' ratings of perceived difficulty, invested effort, and liking for each effort level across both beneficiary conditions.

## Investing effort for others is less motivating than for self in the prosocial effort task

The mean maximum effort level (i.e. the average button-press count across three 6000 ms trials; see details in *Procedure*) was 36.06 (*SD* = 4.95, range = 26.33–46.00; *Figure 2A*).

Participants achieved comparably high success rates in both self-benefiting (*M*=97%) and other-benefiting trials (*M*=96%). A mixed-effects logistic regression model of response success indicated that success rates decreased as effort demands increased (*b*=−4.77, p<0.001). However, no other effects reached significance (ps>0.245; *Supplementary file 1*). Regarding response speed (button presses per second), participants responded faster as required effort increased (*b*=0.56, p<0.001; *Figure 2B*), but their speed did not vary with prospective reward magnitude (*b*=0.01, p=0.128;

*Figure 2C*). Notably, participants also responded faster for themselves than for others (b=–0.07, p<0.001; *Supplementary file 1*).

To validate our experimental manipulation, we examined participants' subjective ratings as a function of effort level and beneficiary using linear mixed-effects regression models. As expected (*Figure 2D*), participants rated higher effort levels as more difficult (b=2.38, p<0.001), more demanding (b=1.90, p<0.001), and less likable (b=–1.79, p<0.001). They felt less effort (b=–0.32, p=0.019) and more disliking (b=–0.62, p=0.001) for other-benefiting trials compared to self-benefiting trials. However, they perceived no differences in task difficulty between self-benefiting and other-benefiting trials (b=0.19, p=0.260). Moreover, the liking rating data exhibited a significant interaction between recipient and effort (b=–0.28, p=0.023). Follow-up simple slopes analyses revealed that the discounting effect of effort on the liking rating was more pronounced when the beneficiary was others (b=–1.93, 95% CI = [–2.28, –1.58], p<0.001) compared to when it was themselves (b=–1.65, 95% CI = [–2.00, –1.30], p<0.001; *Supplementary file 2*). Together, our rating and behavioral data suggest that the effort manipulation was successful. Furthermore, participants were less motivated to invest effort for others than for themselves at both subjective and behavioral levels, despite similar success rates.

## Effort adds reward value for self but discounts reward value for others in the prosocial effort task

In the prosocial effort task, the RewP was evident as a relative positivity over frontocentral areas (*Figure 3*). We fitted the amplitude of the RewP using a linear mixed-effects regression model with recipient, effort, magnitude, valence, and their interactions as predictors. The full regression estimates for RewP data are shown in *Figure 4A*. As predicted, the RewP was significantly larger for self-benefiting relative to other-benefiting trials (b=–0.68, p=0.012) and for gain versus nongain feedback (b=–1.08, p<0.001), and became more positive as reward magnitude increased (b=0.42, p=0.003).

We observed a significant three-way interaction among recipient, magnitude, and valence (b=0.86, p=0.038). Subsequent simple slopes analyses (*Figure 4B*) revealed distinct neural patterns based on the beneficiary. When the beneficiary was self, the RewP became more positive as reward magnitude increased for both gain feedback (b=0.75, 95% CI = [0.31, 1.19], p<0.001) and nongain feedback (b=0.46, 95% CI = [0.02, 0.90], p=0.039). In contrast, when the beneficiary was others, the RewP became more positive as reward magnitude increased for nongain feedback (b=0.51, 95% CI = [0.07, 0.95], p=0.022), but not for gain feedback (b=–0.02, 95% CI = [–0.46, 0.42], p=0.923).

Crucially, we observed a significant two-way interaction between recipient and effort (b=–0.55, p=0.009), which was further qualified by a significant three-way interaction among recipient, effort, and magnitude (b=–0.49, p=0.019). The predicted effects of the three-way interaction among recipient, effort, and magnitude are shown in *Figure 4C*. Visual inspection suggested that when reward magnitude was low, the RewP was insensitive to invested effort across self-benefiting and other-benefiting trials. However, as reward magnitude increased, the effect of effort expenditure on the RewP diverged depending on the beneficiary. To confirm these observations statistically, we conducted post hoc simple slopes analyses at 1 standard deviation (*SD*) below ('Low') and above ('High') the mean reward magnitude. At low reward magnitude, the RewP did not vary with effort level, regardless of whether the beneficiary was self (b=0.01, 95% CI = [–0.42, 0.44], p=0.958) or others (b=–0.05, 95% CI = [–0.48, 0.39], p=0.829). In contrast, a striking dissociation emerged at high reward magnitude. When the beneficiary was self, the RewP tended to be more positive as effort level increased (b=0.42, 95% CI = [–0.01, 0.85], p=0.056), suggesting an effort-enhancement effect. Conversely, when the beneficiary was others, the RewP became less positive as effort level increased (b=–0.63, 95% CI = [–1.05,–0.20], p=0.004), indicating an effort-discounting effect.

To rule out the possibility that the differential vigor between self- and other-benefiting trials drove the Recipient × Effort and Recipient × Effort × Magnitude interactions on the RewP, we conducted two control analyses by including trial-by-trial response speed and subjective effort ratings as separate covariates in the RewP model. Neither response speed (b=–0.07, p=0.614) nor effort rating (b=0.10, p=0.186) predicted RewP amplitudes, and the critical Recipient × Effort and Recipient × Effort × Magnitude interactions remained significant and essentially unchanged (*Supplementary file 3*).

To establish the specificity of our RewP findings, we examined the parietal P3 in response to performance feedback (i.e. effort-completion cues; see *Figure 3—figure supplement 1* for the ERP waveforms and topographic maps) using a linear mixed-effects regression model with recipient,

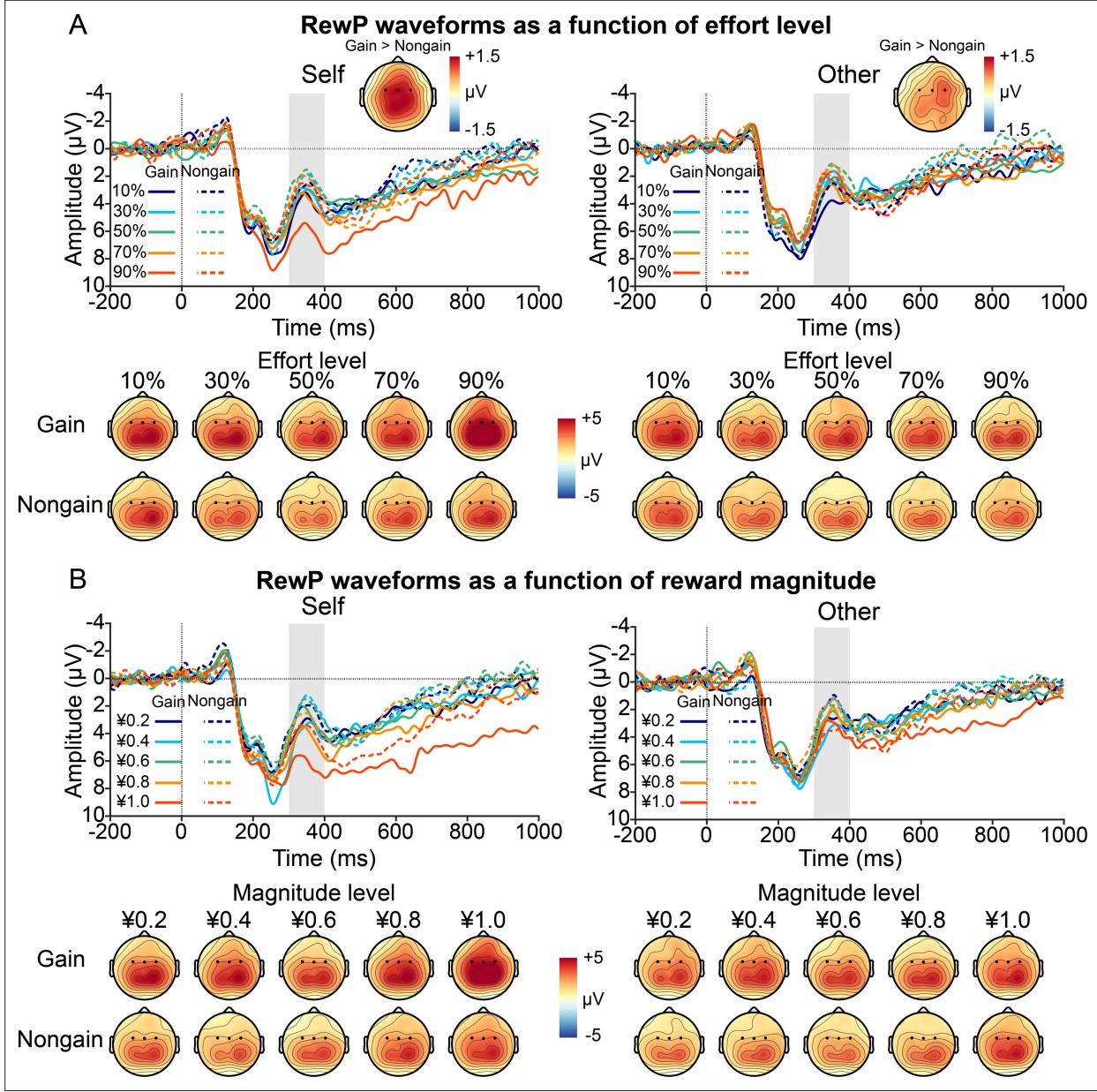

**Figure 3.** Grand-average event-related potential (ERP) waveforms and topographic maps of the reward positivity (RewP) as a function of recipient (self vs. other) and valence (gain vs. nongain) separately for effort (**A**) and reward (**B**) trials. Gray shaded bars represent time windows used for quantification.

The online version of this article includes the following figure supplement(s) for figure 3:

**Figure supplement 1.** Grand-average event-related potential (ERP) waveforms and topographic maps of the P3 as a function of recipient (self vs. other) separately for effort (**A**) and reward (**B**) trials.

effort, magnitude, and their interactions as predictors. We found a main effect of effort on the P3 in response to effort-completion cues, with its amplitudes increasing with prior effort levels ($b=0.72$, $p<0.001$). Importantly, this sensitivity to effort was invariant across beneficiary conditions, as indicated by a nonsignificant interaction between recipient and effort ($b=-0.19$, $p=0.285$). No other significant effects were observed (*Supplementary file 4*).

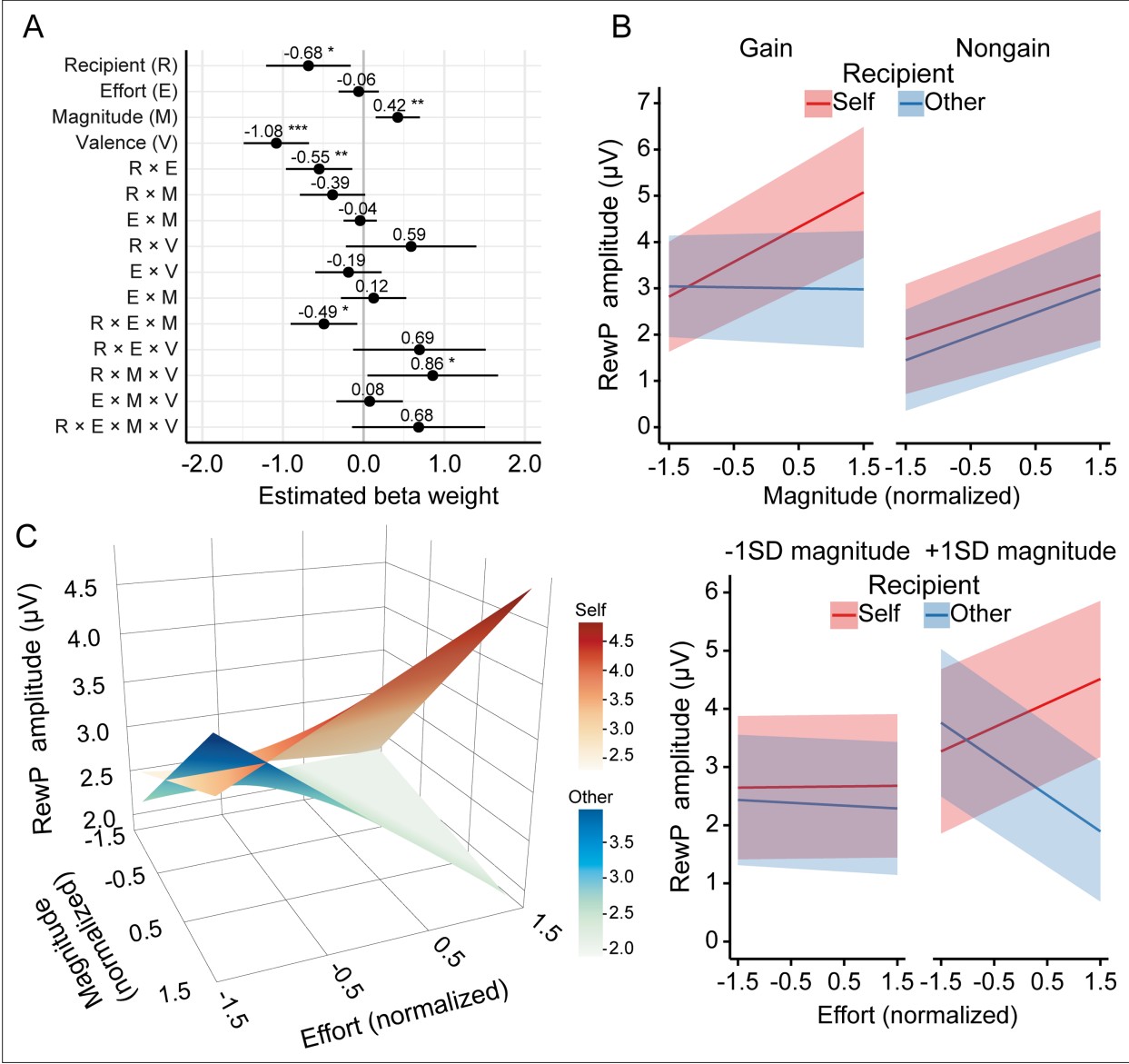

**Figure 4.** Reward positivity (RewP) results in the prosocial effort task. (**A**) Estimated beta weights for the mixed-effects model predicting RewP amplitudes. The RewP model was specified as: Amplitude ~ Recipient * Effort * Magnitude * Valence + (Recipient + Effort + Magnitude | Participant). (**B**) Fixed effects of reward magnitude on the RewP as a function of recipient and valence during reward evaluation, showing a significant three-way interaction. (**C**) Fixed effects of effort and reward on the RewP as a function of recipient during reward evaluation. The left graph displays the fixed effects with effort and reward as continuous predictors, whereas the right graph shows fixed effects of effort at 1 *SD* below and above the mean reward magnitude. An effort-enhancement effect emerged when participants invested effort for themselves, whereas an effort-discounting effect occurred when they exerted effort for others. This dissociable after-effect was present only when reward magnitude was high. Error bars and shaded areas depict 95% confidence intervals (n = 40). *p<0.05, **p<0.01, ***p<0.001.

## Reward is devalued by effort to a higher degree for others than for self in the prosocial decision-making task

Decision times in the prosocial decision-making task (*Figure 5A and B*) were fitted using a linear mixed-effects regression model with recipient, magnitude, and both linear and quadratic terms for effort, along with their interactions, as predictors. The analysis yielded significant main effects of effort (linear: *b*=38.91, p=0.003; quadratic: *b*=−64.89, p<0.001) and magnitude (*b*=−54.88, p<0.001). Crucially, we observed significant interactions between recipient and effort (*b*=−41.49, p=0.001), as well as between recipient and magnitude (*b*=44.86, p=0.020). Simple slopes analyses indicated that decision times increased with effort in self-benefiting trials (*b*=59.70, 95% CI = [32.87, 86.50], p<0.001)

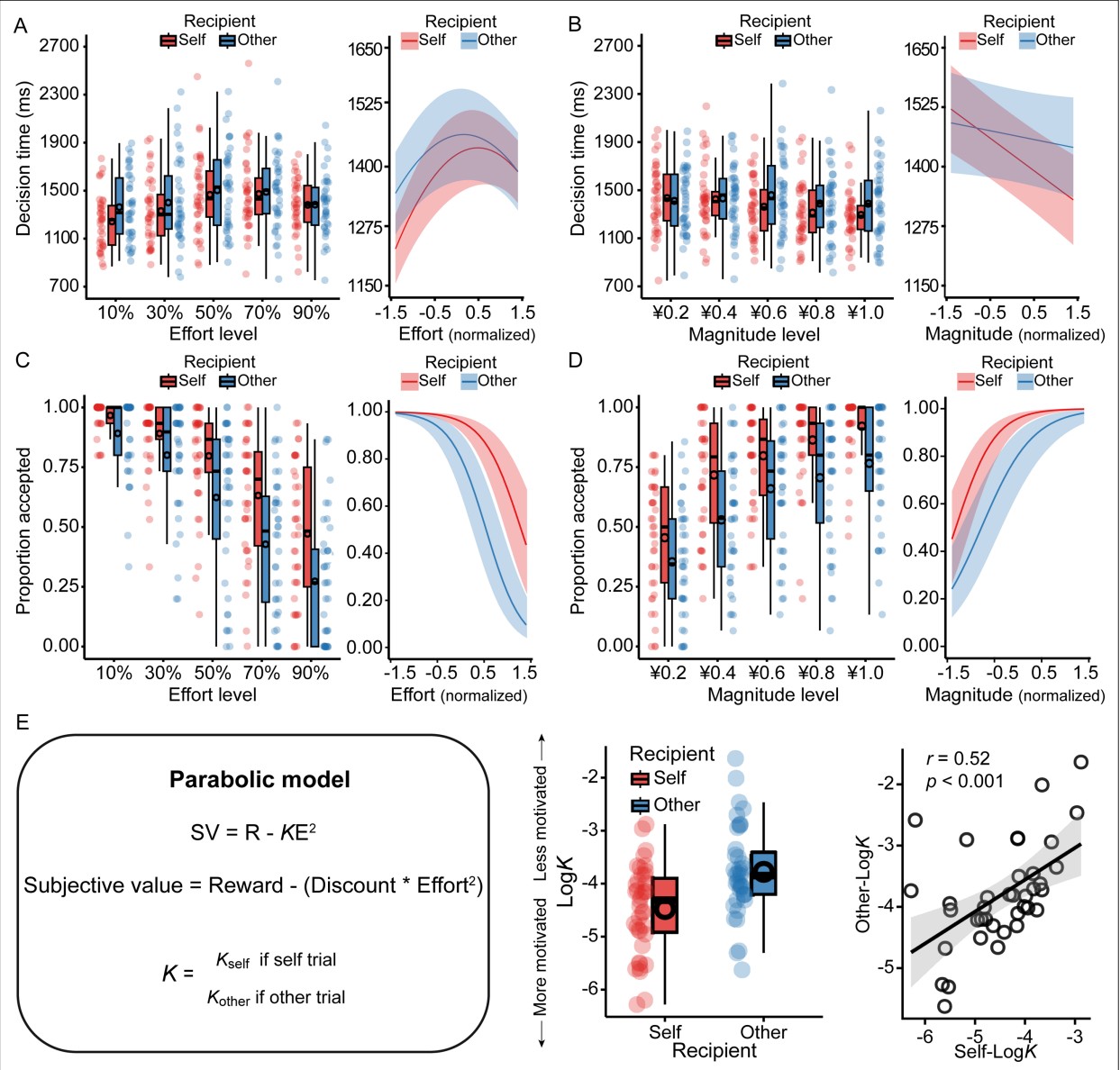

**Figure 5.** Behavioral and discounting results of the prosocial decision-making task. (**A–B**) Participants took longer to make decisions as effort level increased in self-benefiting trials but not in other-benefiting trials (**A**). Increased reward magnitude decreased the decision time more pronouncedly in self-benefiting trials than in other-benefiting trials (**B**). (**C–D**) Participants were less willing to invest effort for others than for themselves. (**E**) Effort exertion discounted rewards to a higher degree when the beneficiary was others compared to when it was themselves (left and middle). A higher discounting rate for others was associated with a higher discounting rate for self (right). Data were analyzed using linear mixed-effects models for panels A–B, mixed-effects logistic models for panels C–D, and Pearson correlation for panel E. For panels A–D, the left panels display the raw data overlaid with boxplots, whereas the right panels show the model-predicted fixed effects. The black circles overlaid on the boxplots indicate the mean across participants. Shaded areas depict 95% confidence intervals. The sample size is n = 40, except for the $K$ analyses in E (middle and right) where n = 38 because two participants had negative original $K$ values and were thus excluded. Note that seven participants had an accuracy rate of less than 60% on catch trials, but this did not influence the results of the prosocial decision-making task.

but not in other-benefiting trials ($b$=18.30, 95% CI = [–8.55, 45.10], p=0.182). Similarly, higher reward magnitude facilitated response speed more substantially in self-benefiting trials ($b$=–77.30, 95% CI = [–105.70, –48.90], p<0.001) than in other-benefiting trials ($b$=–32.50, 95% CI = [–60.80, –4.20], p=0.024). Furthermore, both the linear and quadratic effects of effort were qualified by significant interactions with reward magnitude (linear interaction: $b$=33.23, p<0.001; quadratic interaction: $b$=15.14, p=0.040). Simple slopes analyses of the linear term showed that decision times increased with effort at high reward magnitude (M+1$SD$: $b$=72.10, 95% CI = [45.30, 98.90], p<0.001) but not at

low reward magnitude (*M* − 1*SD*: b=5.61, 95% CI = [−21.20, 32.40], p=0.682). Decomposition of the quadratic interaction revealed that the nonlinear curvature of the effort effect was more pronounced at low reward magnitude (*M* − 1*SD*: b=−160.10, 95% CI = [−218.30, −101.90], p<0.001) than at high reward magnitude (*M*+1*SD*: b=−99.50, 95% CI = [−157.60, −41.40], p=0.001; *Supplementary file 5*).

We fitted participants' choice behavior using a mixed-effects logistic regression model with recipient, effort, magnitude, and their interactions as predictors. As depicted in *Figure 5C and D*, participants were less willing to invest effort as effort levels increased (b=−2.54, p<0.001) and more willing as reward magnitude increased (b=2.11, p<0.001). They were also less willing to exert effort for others than for themselves (b=−1.74, p<0.001). This recipient effect was modified by a significant interaction between recipient and magnitude (b=−0.56, p<0.001). Simple slopes analyses revealed that the incentive effect of reward on willingness to exert effort was less pronounced when the beneficiary was another person (b=1.83, 95% CI = [1.36, 2.31], p<0.001) compared to when it was participants themselves (b=2.39, 95% CI = [1.89, 2.90], p<0.001). We also found a significant interaction between effort and magnitude (b=0.15, p=0.028). Follow-up simple slopes analyses revealed that the discounting effect of effort on willingness was more pronounced at low reward magnitude (*M* − 1*SD*: b=−2.69, 95% CI = [−3.09, −2.29], p<0.001) than at high reward magnitude (*M*+1*SD*: b=−2.38, 95% CI = [−2.82, −1.94], p<0.001; *Supplementary file 6*). These findings were further corroborated by a parabolic discounting model (see Materials and methods). As shown in *Figure 5E*, participants exhibited a higher (i.e. steeper) discounting rate for other-benefiting choices (*M*=−3.78) compared to self-benefiting choices (*M*=−4.48, p<0.001). Furthermore, a higher discounting rate for others was

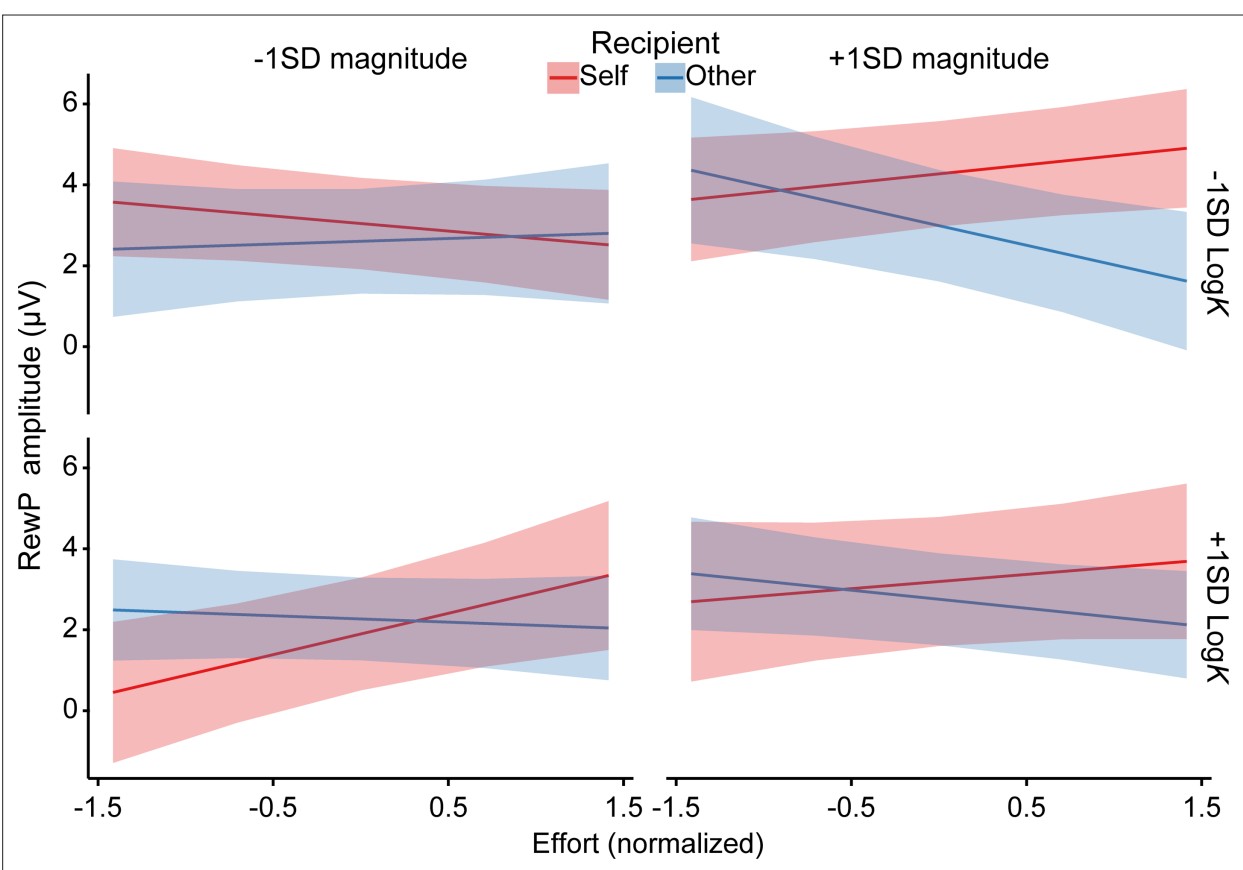

**Figure 6.** Cross-task modulation results. Individual differences in effort discounting (log-transformed *K*) estimated from the prosocial decision-making task modulated the neural after-effects of effort exertion in the prosocial effort task. For self-benefiting trials, high-discounting individuals exhibited an effort-enhancement effect on the reward positivity (RewP) specifically at low reward magnitude; conversely, for other-benefiting trials, low-discounting individuals exhibited an effort-discounting effect only at high reward magnitude. These plots decompose the significant four-way interaction (recipient × effort × magnitude × *K* value) derived from the linear mixed-effects model. Fixed effects are visualized at ±1 *SD* from the mean. Shaded areas depict 95% confidence intervals (n = 40). Note that two participants had negative original *K* values.

associated with a higher discounting rate for oneself ($r$=0.52, p<0.001), suggesting a domain-general motivation that influences effort discounting across beneficiary conditions.

## Neural after-effects of prosocial effort from the prosocial effort task are modulated by effort discounting rate from the prosocial decision-making task

To bridge our neural findings with behavioral preferences, we examined whether individual differences in effort discounting (quantified by the discounting parameter $K$) from the prosocial decision-making task modulated the neural after-effects of effort (as indexed by the RewP) in the prosocial effort task. To this end, we added participants' $K$ values (log-transformed and $z$-scored) to the above-described RewP model as a fixed effect predictor, allowing it to interact with all other predictors of interest (**Supplementary file 7**).

We observed a significant four-way interaction among recipient, effort, magnitude, and $K$ value ($b$=0.58, p=0.013). Crucially, this higher-order interaction was robust, as it was replicated when we replaced individual $K$ values with high-effort choice proportions from the prosocial decision-making task ($b$=–0.58, p=0.014; **Supplementary file 7**). To decompose this complex interaction, we performed simple slopes analyses separately for self- and other-benefiting trials at high and low levels of reward magnitude and discounting rate (±1 $SD$). As shown in **Figure 6**, for self-benefiting trials, the effort-enhancement effect on the RewP was significant only for participants with high discounting rates at low reward magnitude ($b$=1.02, 95% CI = [0.22, 1.82], p=0.012). In contrast, participants with low discounting rates exhibited no significant effort effect ($b$=–0.37, 95% CI = [–0.89, 0.15], p=0.159). At high reward magnitude, simple slopes analyses detected no significant effort effects for either high ($b$=0.35, 95% CI = [–0.44, 1.14], p=0.383) or low ($b$=0.45, 95% CI = [–0.07, 0.97], p=0.093) discounting individuals. For other-benefiting trials, participants with low discounting rates exhibited a significant effort-discounting effect at high reward magnitude ($b$=–0.97, 95% CI = [–1.74, –0.20], p=0.014). In contrast, no significant effort effects were observed for participants with high discounting rates at either high ($b$=–0.45, 95% CI = [–0.97, 0.08], p=0.098) or low ($b$=–0.16, 95% CI = [–0.69, 0.38], p=0.564) reward magnitudes, nor for participants with low discounting rates at low reward magnitude ($b$=0.14, 95% CI = [–0.64, 0.92], p=0.729).

## Discussion

Many prosocial behaviors require effort investment, yet people are often reluctant to exert effort for others' benefit relative to their own. In this study, we examined how effort expenditure influences subsequent reward evaluation during prosocial acts, addressing a critical gap in previous research. We found dissociable reward after-effects of effort exertion between self-benefiting and other-benefiting acts: prior effort potentiated reward evaluation when the beneficiary was oneself but attenuated it when the beneficiary was another person. Moreover, this dissociation occurred specifically when the potential reward magnitude was large and was modulated by individual differences in effort discounting, yet remained independent of performance evaluation.

In our study, despite perceiving no differences in task difficulty and achieving comparable success rates for themselves and others, participants took longer to exert effort, reported less effort, and disliked the task more as the required effort increased when helping others. Participants exhibited a higher discounting rate for choices benefiting others versus those benefiting themselves, and their decision time was less influenced by effort and reward levels when the beneficiary was others compared to when it was themselves. These findings are largely consistent with previous studies, demonstrating a prosocial apathy when physical effort is required to help others (**Contreras-Huerta et al., 2020**).

Our most important finding is the dissociation of the reward after-effect of effort expenditure between self-benefiting and other-benefiting trials. When the beneficiary was oneself, the RewP was potentiated as prior invested effort increased. In contrast, when the beneficiary was others, the RewP was attenuated as prior invested effort increased. Given that the RewP is considered a reliable neural signature of reward sensitivity (**Proudfit, 2015**), our data suggest an effort-enhancement effect for self-benefiting acts but an effort-discounting effect for other-benefiting acts. The effort-enhancement effect aligns with previous studies where effort is exerted to obtain rewards for oneself

(**Bogdanov et al., 2022**; **Ma et al., 2014**; **Umemoto et al., 2023**; **Yi et al., 2020**). Several theories have been proposed to interpret this effect. For instance, prior effort expenditure may enhance the subjective value of a reward through cognitive dissonance reduction (i.e. effort justification) to alleviate the psychological discomfort resulting from having engaged in unpleasant effort (**Aronson and Mills, 1959**) or via the psychological contrast between the aversive state elicited by effort expenditure and the reward that follows (**Zentall, 2010**). While these perspectives adequately explain the effort-enhancement effect for self-benefiting acts, they fall short in explaining the discounting effect observed here for other-benefiting acts. According to these views, the more aversive state elicited by investing effort for others should lead to stronger justification or a more intense psychological contrast, which consequently should increase the reward value to a greater degree. However, we observed an effort-discounting effect when it came to prosocial acts. This discounting effect may result from the heightened salience of opportunity costs in helping others, i.e., the value of the next-best use of the effort devoted to the current task (**Kurzban et al., 2013**).

Interestingly, the dissociable after-effect of effort expenditure occurred only when reward magnitude was high. When reward magnitude was low, prior invested effort did not affect the RewP across self-benefiting and other-benefiting trials. These results suggest that the dissociable effort after-effect relies on the involvement of the motivational system, which might be activated only when potential reward is high. This possibility was further supported by our P3 data in response to effort-completion cues during performance evaluation. Specifically, participants exhibited an increased P3 when seeing a feedback stimulus informing them that the required effort has been achieved, whether the beneficiary was themselves or others. As the P3 is thought to reflect motivational salience based on feedback evaluation (**Nieuwenhuis et al., 2005**), our results suggest that participants could derive value from their successful effort completion independently of the reward outcome. Unlike reward evaluation, the effort-enhancement effect on the P3 is associated with intrinsic motivation such as pride and achievement stemming from successful performance (**Bowyer et al., 2021**; **Jiang and Zheng, 2023**). Together, our data establish that the dissociation in the after-effect of prosocial effort occurs only when reward motivation is sufficiently activated.

While previous research on prosocial effort has focused exclusively on cognitive processes before and during effort expenditure (**Contreras-Huerta et al., 2020**), our findings provide a comprehensive picture of prosocial effort. We not only replicated the greater effort discounting for prosocial acts compared to self-benefiting acts but also demonstrated a dissociable after-effect of prosocial effort. Our results suggest that the difference between self-benefiting and other-benefiting effort is quantitative before and during effort expenditure but qualitative after effort expenditure. This after-effect of prosocial effort, together with the established discounting effect before effort exertion, offers a new perspective for facilitating prosocial behavior. When working for oneself, effort expenditure not only prospectively discounts but also retrospectively increases the subjective value of reward, a phenomenon referred to as the *effort paradox* (**Inzlicht et al., 2018**). Crucially, our cross-task analysis reveals that this paradox is not universal but is driven specifically by individuals highly sensitive to effort costs (high $K$), particularly when external rewards are low. This finding strongly supports a cognitive dissonance account: those who find effort most aversive are most compelled to inflate the value of small rewards to justify their exertion (**Aronson and Mills, 1959**).

In contrast, for prosocial acts, this justification mechanism appears absent. Instead, we observed a persistent effort discounting before, during, and after effort expenditure, which was most pronounced in individuals with low effort sensitivity (low $K$) when reward magnitude was high. This seemingly paradoxical pattern might be interpreted through the lens of disadvantageous inequity aversion (**Fehr and Schmidt, 1999**). Specifically, the combination of high personal effort and high monetary reward for another person creates a salient disparity between the participant's incurred cost and the recipient's gain. Although low-$K$ individuals are behaviorally willing to tolerate this cost, their neural valuation system may nonetheless track the 'unfairness' of this asymmetry, thereby attenuating the neural reward signal (**Tricomi et al., 2010**). These insights suggest that facilitating prosocial behavior may require not just lowering costs, but potentially framing outcomes to trigger the effort justification mechanisms that drive the effort paradox observed in self-benefiting acts (**Inzlicht and Campbell, 2022**). Consistent with this view, a promising direction for future studies is to manipulate the social distance of the recipient. This approach will help determine whether the after-effect of effort expenditure could shift from an effort-discounting effect to an effort-enhancement effect when the beneficiary

is a close other, such as a friend or family member, thereby bridging the gap between self- and other-oriented motivation (*Jones and Rachlin, 2006*).

One limitation of this study concerns the potential confound between temporal delay and effort level. Because higher effort levels required more time to complete, effort expenditure might not have directly affected subsequent reward evaluation but instead increased participants' delay discounting. If this were the case, one would expect a similar discounting effect on the RewP for self-benefiting trials, as the RewP amplitude typically decreases as the time to receive a reward increases (*Zheng et al., 2023*). Moreover, temporal delay is unlikely to account for our findings given the relatively small difference in duration between the lowest and highest effort levels in our task (*Weinberg et al., 2012*). Nonetheless, future work is needed to strictly control for duration to rule out this possibility. Another concern is that participants exhibited less vigor when working for others, as indicated by slower response speed and lower subjective effort ratings for other- versus self-benefiting trials. Although our control analyses confirmed that neither covariate predicted RewP amplitudes and the critical interactions remained significant, covariates may not fully capture the effects of differential motivation, and this alternative explanation cannot be entirely ruled out.

In conclusion, our results demonstrate that the neural evaluation of reward outcomes is dynamically shaped by the prior cost of effort in a beneficiary-dependent manner. We identified a critical qualitative dissociation: whereas self-benefiting effort paradoxically enhances reward valuation, other-benefiting effort induces a persistent reward devaluation. Crucially, this divergence represents a distinct valuation process that emerges specifically at high reward magnitude, is modulated by individual differences in effort discounting, and operates independently of performance evaluation. These findings imply that the *effort paradox* where labor adds value is restricted to self-benefiting acts. Consequently, promoting prosocial behavior may require not just reducing physical costs, but reframing prosocial outcomes to trigger the justification mechanisms that naturally support self-benefiting exertion.

## Materials and methods
### Participants
Forty-seven right-handed university students were recruited for this study through local advertisements. Seven participants were excluded from data analysis due to the following reasons: five never chose to work on either self-benefiting or other-benefiting trials, one had a success rate of 35% on the maximum effort level during the prosocial effort task, and one doubted the impact of their actions on the other participant. The final sample consisted of 40 participants (20 females; $M$=21.55 years, $SD$ = 2.65). To characterize the statistical sensitivity of our design, we conducted a simulation-based sensitivity analysis on the RewP model using the *simr* v1.0.7 package (*Green and MacLeod, 2016*). For each fixed effect in the linear mixed-effects model, we determined the minimum smallest detectable effect size (unstandardized regression coefficient) at 80% power ($\alpha$=0.05), given the current sample size, number of trials, and variance components estimated from the fitted model (*Supplementary file 8*). All participants had normal or corrected-to-normal vision and reported no psychiatric or neurological disorders. They received ¥25 for participation and a bonus of ¥29–¥37 based on their task performance. Each participant provided written informed consent, and this study was approved by the Institutional Review Board of Guangzhou University (approval number: 2024055).

### Procedure
Upon arrival at the lab, participants undertook a role assignment task, leading them to believe that they would complete two EEG tasks with a partner. Afterward, they performed a prosocial effort task and a prosocial decision-making task while their EEG was recorded. Following the EEG tasks, participants used a 9-point Likert scale (ranging from 1=not at all to 9=very much) to rate their perceived difficulty, invested effort, and liking at each effort level when exerting effort for themselves and others during the tasks.

#### The role assignment task
This task was adapted from a previous study (*Lockwood et al., 2017*). Participants were led to one side of a door and informed that a second participant (who was in fact a confederate) was also involved in this study (*Figure 1A*). The confederate, of the same gender as the participant, was then escorted by

a different experimenter to the opposite side of the door. Both the participant and confederate were handed a black glove and instructed not to speak for anonymity. They acknowledged each other's presence by waving their gloved hands, without ever being seen by each other. The experimenter then tossed a coin to determine who would choose a ball from a box first. Their roles (a receiver vs. a decider) were assigned based on the outcome. Unbeknownst to participants, they were always designated as the decider, responsible for performing tasks for themselves and others. The confederate was assigned the role of the receiver, responsible solely for completing tasks for themselves. To prevent potential effects of social norms such as reciprocity (*Gintis et al., 2003*), both the participant and confederate were informed that they would not be aware of each other's performance and would leave the lab at different times.

## The prosocial effort task

This task was designed to measure participants' neural responses to rewards obtained after investing physical effort for themselves and others (*Figure 1B*). Before the task, participants completed three trials in which they were asked to press a button as rapidly as possible with their nondominant pinky finger for 6000 ms. The maximum effort level was operationalized as the average button-press count across the three trials. In the prosocial effort task, each trial began with a cue for 1500 ms, indicating whether it was self-benefiting or other-benefiting. Participants exerted physical effort to earn rewards for themselves in self-benefiting trials and for others in other-benefiting trials. Self and other trials were highlighted in different colors (blue and red) throughout the trial, which was counterbalanced across participants. After a jittered interval (900–1100 ms), a pie chart with a number below it appeared for 1500 ms, showing the required effort level (10%, 30%, 50%, 70%, or 90% of their maximum effort level) and the potential reward (¥0.2, ¥0.4, ¥0.6, ¥0.8, or ¥1.0), respectively (*Figure 1C*). The five effort levels were fully crossed with the five reward magnitudes, creating 25 unique combinations. Following another jittered interval (900–1100 ms), participants entered the effort-execution phase, making the required button presses with their nondominant pinky finger within 6000 ms. After this phase and another jittered interval (900–1100 ms), a feedback stimulus was shown for 1000 ms, indicating whether the required effort level was achieved. If participants succeeded, a tick was presented, signaling that they were eligible to win the cued reward. If they failed, a cross was displayed, followed by a new trial after a jittered interval of 900–1100 ms. After a 2500 ms interval, a feedback stimulus was displayed for 1000 ms, indicating whether participants received the reward or not. Gains and nongains were equally likely and delivered pseudorandomly. Each trial ended with an interval varying between 900 and 1100 ms. The task consisted of 200 trials (100 for self-benefiting and 100 for other-benefiting, respectively) divided into 8 blocks of 25 trials, with a self-determined break between blocks. Eight practice trials familiarized participants with the task before the formal experiment.

## The prosocial decision-making task

This task was designed to measure participants' willingness to exert effort for themselves and others (*Figure 1D*). Participants made decisions between a baseline no-effort option for ¥0.1 and a high-effort option for a greater reward. The high-effort options were the same as those in the prosocial effort task, including 25 unique effort-reward combinations. Participants had 3500 ms to decide using their left or right index finger. The chosen option was highlighted with a green border for 300 ms. Failure to respond within the time limit resulted in ¥0 and a 1000 ms warning message of 'Please respond within 3500 ms'. Each trial ended with a jittered interval of 900–1100 ms. The task consisted of 150 trials, with each high-effort option occurring six times. Half of the trials benefited participants themselves, while the other half benefited others. To ensure that participants stayed focused on the task, we included 20 catch trials (10 for each beneficiary condition) where participants confirmed the effort and reward levels of the previous high-effort option. To ensure that our task was incentive-compatible, participants were told that they would complete their chosen effort option on 16 randomly selected trials (8 for each beneficiary condition) to determine their final reimbursement. They were instructed to consider each decision carefully because each trial choice would be selected. Before the experiment, participants completed 10 practice trials for familiarization. After the task, participants exerted their effort required for their choices in the 16 selected trials. Finally, they were asked to report whether they believed they were earning rewards for the other participant (i.e. the receiver).

## EEG recording and processing

EEG data were recorded using 28 Ag/AgCl channels placed on an elastic cap based on the international 10–20 system. Two additional channels were positioned on the left and right mastoids. The EEG were recorded with a reference channel placed between Cz and CPz. Horizontal and vertical electrooculograms were recorded from two pairs of channels over the external canthi of each eye and the left suborbital and supraorbital ridges, respectively. EEG signals were amplified using a Neuroscan Grael 4K amplifier with a low-pass filter of 100 Hz in DC acquisition mode and digitized at a rate of 512 samples per second. Channel impedances were maintained below 5 kΩ.

The EEG data were analyzed using EEGLAB v2021.0 (*Delorme and Makeig, 2004*) and ERPLAB v8.10 (*Lopez-Calderon and Luck, 2014*) toolboxes in MATLAB 2020b (MathWorks, USA). The signals were rereferenced to the average of the left and right mastoids and filtered with a bandpass of 0.1–35 Hz using a zero phase-shift Butterworth filter (12 dB/octave roll-off). Channels with poor quality or excessive noise were interpolated using the spherical interpolation algorithm, and portions of EEG containing extreme voltage offsets or break periods were removed. Ocular artifacts were removed using an Infomax independent component analysis on continuous EEG with the help of the ICLabel algorithm (*Pion-Tonachini et al., 2019*). Epochs were then extracted from –200 to 1000 ms relative to feedback onset, with the prestimulus average activity as the baseline. An automatic artifact detection algorithm was applied to remove epochs with a voltage difference exceeding 50 µV between sample points or 200 µV within a trial, a maximum voltage difference less than 0.5 µV within 100 ms intervals, or a slow voltage drift with a slope greater than ±100 µV. On average, 97.71% of trials were retained for statistical analysis. Single-trial RewP amplitude was measured as mean voltage from 300 to 400 ms relative to reward feedback onset (i.e. reward delivery) over frontocentral channels (FC3, FCz, FC4). We also measured the parietal P3 (300–440 ms; averaged across P3, Pz, and P4) in response to performance feedback (i.e. effort completion), given its relationship with motivational salience (*Bowyer et al., 2021*; *Ma et al., 2014*). Measurement parameters (time windows and channel sites) were determined from the grand-averaged ERP waveforms and topographic maps collapsed across all conditions, which was thus orthogonal to the conditions of interest (*Luck and Gaspelin, 2017*). Data were averaged across the selected electrode clusters to improve signal-to-noise ratio and reliability.

## Data analysis

Single-trial data were exported into R v4.2.2 for statistical analyses. Key statistical analyses utilized mixed-effects regression models with random intercepts and slopes (unstructured covariance matrix), implemented in the *lme4* package v1.1.31 (*Bates et al., 2015*). For the prosocial effort task, we analyzed response success data using a mixed-effects logistic regression model. We analyzed response speed (i.e. button presses per second) and ERP data using linear mixed-effects regression models. These models included recipient, effort, magnitude, valence, and their interactions as predictors. For the prosocial decision-making task, we fitted decision time data using a linear mixed-effects regression model and choice data using a mixed-effects logistic regression model. The choice model included recipient, effort, magnitude, and their interactions as predictors. For the decision time model, visual inspection of the data (*Figure 5A*, left panel) suggested a nonlinear relationship between effort and decision time. Therefore, we included both linear and quadratic terms for effort, along with recipient, magnitude, and their interactions, as predictors. We fitted post-experimental rating data separately using a linear mixed-effects regression model with recipient, effort, and their interactions as predictors. For all models, we contrast-coded categorical regressors (recipient: –0.5 for self and +0.5 for other; valence: –0.5 for gain and +0.5 for nongain) and z-scored continuous regressors (effort and magnitude) within participants. For each model, we fitted the maximal random-effects structure and, when the model was overparameterized, used singular value decomposition to simplify the random-effects structure until the model converged. For linear mixed-effects models, p-values were calculated using Satterthwaite's approximation for degrees of freedom. For mixed-effects logistic models, p-values were obtained using Wald Z-tests. To decompose significant interaction, we performed follow-up pairwise comparisons on estimated marginal means. We excluded trials with failed responses (3.34%) in the prosocial effort task and trials with no responses (0.91%) in the prosocial decision-making task from statistical analyses.

To quantify how participants devalued rewards by effort exertion for themselves and others, we fitted their choices in the prosocial decision-making task separately for self-benefiting and other-benefiting trials with a parabolic function:

$$SV = R - KE^2$$

where *SV* represents the subjective net value of the high-effort option with a given reward (*R*) and effort (*E*). The discount parameter *K* characterizes the degree to which the reward is discounted by required effort. A higher *K* value indicates that the reward is devalued by the effort to a higher degree. The derived *SV* of the high-effort and baseline no-effort options were compared and transformed into the probability of choosing the high-effort option through a softmax function:

$$P_{high\,effort} = \frac{e^{\beta * SV_{high\,effort}}}{e^{\beta * SV_{no\,effort}} + e^{\beta * SV_{high\,effort}}}$$

where $\beta$ represents the slope of the logistic function, which is a participant-specific parameter and reflects the sensitivity to *SV* differences between options (*Collins and Shenhav, 2022*). The parabolic model with two separate *K* parameters for self and other trials has been validated in previous studies using a similar task (*Lockwood et al., 2021*; *Lockwood et al., 2022*). We normalized the *K* values by taking the natural logarithm (i.e. log*K*) and compared these log*K* values between self and other trials using a paired-*t* test.

## Acknowledgements

This work was supported by the National Natural Science Foundation of China (32571255 and 31971027) and the 2024 Tertiary Education Scientific Research Project of Guangzhou Municipal Education Bureau (2024312195).

## Additional information

### Funding

| Funder | Grant reference number | Author |
|---|---|---|
| National Natural Science Foundation of China | 32571255 | Ya Zheng |
| National Natural Science Foundation of China | 31971027 | Ya Zheng |
| 2024 Tertiary Education Scientific Research Project of Guangzhou Municipal Education Bureau | 2024312195 | Ya Zheng |

The funders had no role in study design, data collection and interpretation, or the decision to submit the work for publication.

### Author contributions

Ya Zheng, Conceptualization, Resources, Supervision, Funding acquisition, Validation, Investigation, Methodology, Writing – original draft, Project administration, Writing – review and editing; Rumeng Tang, Data curation, Formal analysis, Validation, Investigation, Visualization, Writing – original draft

### Author ORCIDs

Ya Zheng  https://orcid.org/0000-0002-7469-5955
Rumeng Tang  https://orcid.org/0009-0005-5243-5772

### Ethics

Human subjects: All participants provided written informed consent prior to participation. The study was approved by the Institutional Review Board of Guangzhou University (approval number: 2024055).

Reviewer #1 (Public review): https://doi.org/10.7554/eLife.103566.4.sa1
Reviewer #2 (Public review): https://doi.org/10.7554/eLife.103566.4.sa2
Author response https://doi.org/10.7554/eLife.103566.4.sa3

## Additional files

### Supplementary files

Supplementary file 1. Results of mixed-effects models predicting response success (logistic; left) and response speed (linear; right) in the prosocial effort task.

Supplementary file 2. Results of linear regression models predicting rating data of difficulty, effort, and liking.

Supplementary file 3. Results of reward positivity (RewP) models with response speed (left) and effort rating (right) as covariates.

Supplementary file 4. Results of a linear mixed-effects model predicting P3 amplitudes in response to performance feedback in the prosocial effort task.

Supplementary file 5. Results of a linear mixed-effects model predicting decision times in the prosocial decision-making task.

Supplementary file 6. Results of a mixed-effects logistic regression model predicting decision choices in the prosocial decision-making task.

Supplementary file 7. Results of reward positivity (RewP) models with discounting rate ($\log K$) and high-effort choice proportions as fixed predictors.

Supplementary file 8. Simulation-based sensitivity analysis for fixed effects in the reward positivity (RewP) model.

MDAR checklist

### Data availability

Data and code that support the findings of this study are available on Open Science Framework at https://osf.io/bvpa2/.

The following dataset was generated:

| Author(s) | Year | Dataset title | Dataset URL | Database and Identifier |
|---|---|---|---|---|
| Zheng Y | 2026 | Effort produces after-effects costly for others but valued for self | https://osf.io/bvpa2/ | Open Science Framework, bvpa2 |

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
