## [Editor Report · eLife Assessment]

The findings in this paper provide **solid** support for a hypothesis that has **valuable** implications at the intersection of value-based and social decision-making. The findings suggest that the brain processes rewards received for effort differently when they are earned for themselves versus someone else.

---

## [Referee Report · Reviewer #1 (Public review)]

[Editors' note: this version has been assessed by the Reviewing Editor without further input from the original reviewers. The authors have addressed the comments raised in the previous round of review.]

Summary:

The Authors test the hypotheses, using and effort-exertion and an effort-based decision-making task, while recording brain dynamics with EEG, that the brain processes reward outcomes for effort differentially when they earned for themselves versus others.

Strengths:

The strengths of this experiment include what appears to be a novel finding of opposite signed effects of effort on the processing of reward outcomes when the recipient is self versus others. Also, the experiment is well-designed, the study seems sufficiently powered, and the data and code are publicly available.

Weaknesses:

There is some concern about the fact that participants report feeling less subjective effort, but also more disliking of tasks when they were earning rewards for others versus self. The concern is that participants worked with less vigor during self-versus-others trials and this may partly account for a key two-way Recipient x Effort interaction on the size of the Reward Positivity EEG component. Of note, participants took longer to complete tasks when working for others. While it is true that, in all cases, participants met the requisite task demands (they pressed the required number of buttons) they did so more sluggishly when earning rewards for others. The Authors argue that this reflects less motivation when working for others, which is a plausible explanation. The Authors also try to rule out this diminished vigor as a confounding explanation by showing that the two way interaction remains even when including reaction times (and also self-reported task liking) as a covariate. Nevertheless, it is possible that covariates do not fully account for the effects of differential motivation levels which would otherwise explain the two-way interaction. As such, I think a caveat is warranted regarding this particular result.

---

## [Referee Report · Reviewer #2 (Public review)]

Summary:

Measurements of the reward positivity, an electrophysiological component elicited during reward evaluation, have previously been used to understand how self-benefitting effort expenditure influences processing of rewards. The present study is the first to complement those measurements with electrophysiological reward after-effects of effort expenditure during prosocial acts. The results provide solid evidence that effort adds reward value when the recipient of the reward is the self but discounts reward value when the beneficiary is another individual.

Strengths:

An important strength of the study is that amount of effort, the prospective reward, the recipient of the reward, and whether the reward was actually gained or not were parametrically and orthogonally varied. In addition, the researchers examined whether the pattern of results generalized to decisions about future efforts. The sample size (N=40) and mixed-effects regression models are also appropriate for addressing the key research questions. Those conclusions are plausible and adequately supported by statistical analyses.

---

## [Author Response]

The following is the authors’ response to the previous reviews

**Public Reviews:**

**Reviewer #1 (Public review):**
Summary:The authors test the hypotheses, using an effort-exertion and an effort-based decision-making task, while recording brain dynamics with EEG, that the brain processes reward outcomes for effort differentially when they earned for themselves versus others.Strengths:The strengths of this experiment include what appears to be a novel finding of opposite signed effects of effort on the processing of reward outcomes when the recipient is self versus others. Also, the experiment is well-designed, the study seems sufficiently powered, and the data and code are publicly available.Weaknesses:There is some concern about the fact that participants report feeling less subjective effort, but also more disliking of tasks when they were earning rewards for others versus self. The concern is that participants worked with less vigor during self-versus-others trials and this may partly account for a key two-way Recipient x Effort interaction on the size of the Reward Positivity EEG component. Of note, participants took longer to complete tasks when working for others. While it is true that, in all cases, participants met the requisite task demands (they pressed the required number of buttons) they did so more sluggishly when earning rewards for others. The Authors argue that this reflects less motivation when working for others, which is a plausible explanation. The Authors also try to rule out this diminished vigor as a confounding explanation by showing that the two way interaction remains even when including reaction times (and also self-reported task liking) as a covariate. Nevertheless, it is possible that covariates do not fully account for the effects of differential motivation levels which would otherwise explain the two-way interaction. As such, I think a caveat is warranted regarding this particular result.

We thank Reviewer #1 for the continued positive assessment and for continuing to highlight the caveat regarding the potential influence of differential vigor on the observed RewP interaction effects.

We agree that a caveat is warranted. As detailed in our previous response (R5), we had already conducted control analyses addressing this concern; however, we acknowledge that these results were not incorporated into the manuscript itself. We have now addressed this by adding the covariate analyses to the Result section, along with an explicit caveat in the Discussion.

Before describing the specific revisions, we would like to offer a minor clarification: the covariates in our control analyses were trial-by-trial response speed and self-reported effort ratings, rather than task liking ratings as noted in the summary above. Neither response speed nor effort rating predicted RewP amplitudes, and the critical Recipient × Effort and Recipient × Effort × Magnitude interactions remained significant and essentially unchanged. However, as the reviewer rightly pointed out, covariates may not fully capture the effects of differential motivation. Specifically, we have made the following revisions:

First, we added the covariate control analyses to the Result section: “To rule out the possibility that the differential vigor between self- and other-benefiting trials drove the Recipient × Effort and Recipient × Effort × Magnitude interactions on the RewP, we conducted two control analyses by including trial-by-trial response speed and subjective effort ratings as separate covariates in the RewP model. Neither response speed (*b* = -0.07, *p* = .641) nor effort rating (*b* = 0.10, *p* = .186) predicted RewP amplitudes, and the critical Recipient × Effort and Recipient × Effort × Magnitude interactions remained significant and essentially unchanged (see Supplementary Table S3 for full regression estimates)” (page 12, para. 1).

Second, we added a caveat to the Discussion section acknowledging this alterative explanation, which reads, “Another concern is that participants exhibited less vigor when working for others, as indicated by slower response speed and lower subjective effort ratings for other- versus self-benefiting trials. Although our control analyses confirmed that neither covariate predicted RewP amplitudes and the critical interactions remained significant, covariates may not fully capture the effects of differential motivation, and this alternative explanation cannot be entirely ruled out” (page 22, para. 2, lines 9–12; page 23, para. 1).

**Reviewer #2 (Public review):**
Summary:Measurements of the reward positivity, an electrophysiological component elicited during reward evaluation, have previously been used to understand how self-benefitting effort expenditure influences processing of rewards. The present study is the first to complement those measurements with electrophysiological reward after-effects of effort expenditure during prosocial acts. The results provide solid evidence that effort adds reward value when the recipient of the reward is the self but discounts reward value when the beneficiary is another individual.Strengths:An important strength of the study is that amount of effort, the prospective reward, the recipient of the reward, and whether the reward was actually gained or not were parametrically and orthogonally varied. In addition, the researchers examined whether the pattern of results generalized to decisions about future efforts. The sample size (N=40) and mixed-effects regression models are also appropriate for addressing the key research questions. Those conclusions are plausible and adequately supported by statistical analyses.

We sincerely appreciate Reviewer #2’s positive evaluation of our manuscript and thank the reviewer for recognizing the strength of our experimental design and analysis approach.